# Practice, governance, and culture characteristics of lived experience organisations, and evidence of efficacy: A scoping review protocol

**Jessica E. Opie**[1,2]*, **An Vuong**[1,2], **Alexandra Macafee**[3], **Hanan Khalil**[2], **Natalie Pearce**[4], **Erandathie Jayakody**[3], **Christropher Maylea**[2], **Jennifer E. McIntosh**[1,2]

**1** Bouverie Centre La Trobe University, Melbourne, Victoria, Australia, **2** La Trobe University, Melbourne, Victoria, Australia, **3** The Victorian Mental Illness Awareness Council (VMIAC), Melbourne, Victoria, Australia, **4** Library, Latrobe University, Bendigo, Victoria, Australia

* j.opie@latrobe.edu.au

**Data Availability Statement:** No datasets were generated or analysed during the current study. All

## Abstract

### Background

Mental health policy and service design is increasingly recognizing the importance of the lived experience voice and its inclusion in all aspects of work. Effective inclusion requires a deeper understanding of how best to support lived experience workforce and community members to meaningfully participate in the system.

### Objectives

This scoping review aims to identify key features of organizational practice and governance that facilitate the safe inclusion of lived experience in decision-making and practice within mental health sector contexts. Specifically, the review focuses on mental health organizations devoted to lived experience advocacy or peer support or those in which lived experience membership (paid or voluntary) is central to advocacy and peer support operations.

### Methods

This review protocol was prepared with the Preferred Reporting Items for Systematic Review and Meta-Analysis Protocols and registered with the Open Science Framework. The review will be guided by the Joanna Briggs Institute methodology framework and is being conducted by a multidisciplinary team including lived experience research fellows. It will include published and grey literature, including government reports, organizational online documents, and theses. Included studies will be identified through comprehensive searches of five databases: PsycINFO (Ovid), CINAHL (EBSCO), EMBASE (Ovid), MEDLINE (Ovid), and ProQuest Central. Studies published in English from 2000 onwards will be included. Data extraction will be guided by pre-determined extraction instruments. Results will be presented in a Preferred Reporting Items for Systematic Reviews and Meta-Analyses extension for Scoping Reviews flow chart. Results will be presented in tabular form and

relevant data from this study will be made available upon study completion.

**Funding:** This scoping review is conducted with funding from the Victorian Department of Health and the Victorian Mental Illness Awareness Council (VMIAC). VMIAC lived experience staff had and will have a role in study design, data collection and analysis, decision to publish, and preparation of the manuscript. This collaboration was formed in order to mentoring a lived experience researcher officer who is based at VMIAC.

**Competing interests:** The authors have declared that no competing interests exist.

narratively synthesized. The planned commencement and completion dates for this review were July 1, 2022 and April 1, 2023.

## Discussion

It is anticipated that this scoping review will map the current evidence base underpinning organizational practices in which lived experience workers are involved, specifically in the mental health system. It will also inform future mental health policy and research.

## Trial registration

**Registration:** Open Science Framework (registered: July 26, 2022; registration DOI: 10.17605/OSF.IO/NB3S5).

## Introduction

Trends in mental health policy and practice in Australia and internationally have increasingly acknowledged and prioritized the importance of consumer participation in service provision and design [1]. Despite a public policy priority of lived experience inclusion, structural barriers and power imbalances currently limit meaningful collaboration and effective participation for consumers [2]. Thus, there remains a continual and pressing need for consumer workforce development, including improved career pathways, support and supervision, and the leadership of lived experience workers [3]. Presently, consumers are at a disadvantage both as service users and in the mental health sector workforce, and successful reform is dependent on these challenges being mitigated and the lived experience voice having a central and meaningful role in operations and governance.

Consumers have their own individual experience of mental health challenges, recovery, and using mental health services. As such, they also represent a collective and unique discipline of mental health known as 'consumer perspective' in the Australian context [4]. In Australia, the mental health system is attempting to realign its processes in line with a rapidly evolving focus on consumer informed care and respect for human rights, to address widespread experiences of consumer disempowerment and disadvantage [5]. Related failings of the system to date have been brought into sharp relief by sector reviews, such as the Victorian Royal Commission in Mental Health [6], and include deeply embedded systemic problems entrenched in existing organization structures and norms that have dictated service design and delivery to date. In turn, solutions at the whole of systems-level are recommended, placing the consumer voice and consumer leadership at the centre of reform [6]. With this comes a need to develop the roles and opportunities for effective participation and leadership of people with lived experience, supported by a new government agency led by consumers [5]. While these opportunities are emerging across the mental health sector, demand for consumer participation is accelerating at a rate that is not yet fully supported by the capacity of the lived experience workforce and the readiness of sector organizations.

Barriers to inclusion of consumers in decision-making include inherent power imbalances within the mental health sector, and limited trust between consumer and clinicians and policymakers [7]. Victoria's lived experience engagement framework stresses the importance of consumer participation activities addressing safety and power–recognising that consumers have often experienced significant powerlessness in their interactions with the mental health system

and that being in decision-making spaces can be intimidating and disempowering [8]. The existing culture of traditional workplaces is often alienating for consumers, with lived experience workers often lacking confidence due to a fear of being seen as unprofessional in spaces where consumer values are not entrenched in the organization, particularly at the executive management level [9]. Organizational readiness and commitment are critical to supporting and empowering lived experience workers, with best practice examples involving organizations embracing long term organizational and cultural change influenced by lived experience values [9]. Strategies for addressing the imbalance of power and empowering consumers are discussed in literature on co-production and characterised as "endless" [4]. Power re-distribution and supporting consumers is a complex undertaking that does not lend itself to single prescriptive measures, and thus further research and consolidation of that work is critical in establishing a code of best practice and organizational framework.

Effective engagement of consumer workers and leaders in the mental health system is dependent on understanding the optimal support and practice structures that promote the safe inclusion of expertise from those with lived experience. Approaches to consumer participation risk being tokenistic–seeking support after decisions have been made, relying on a single lived experience representative to advocate for an entire community and discipline, and providing limited career opportunities and remuneration for lived experience work [5]. Creation of sustainable opportunities for leadership for consumers would enhance respect for this unique knowledge base, and likely decrease stigmatization of mental illness and distress that encourage consumers to be transparent about their lived experience and to more confidently take on positions of leadership [6].

Within a fast-paced and ambitious reform agenda in Australia, government and organizations must be able to access comprehensive strategies for facilitating lived experience leadership in an evolving system. Professional training in areas of strategic and technical expertise within well considered governance and support structures is needed, yet best practice guidelines to date do not exist [10].

The primary purpose of the current review is to contribute to this new knowledge base, bringing academic and consumer voices together to consider how an organization can effectively support all levels of consumer workers and act as a voice for its lived experience membership and community members. Specifically, the review methodology and synthesis will be informed by consultation with the Victorian Mental Illness Awareness Council (VMIAC), the peak consumer body in Victoria and a consumer-run and led organization. As the lived experience workforce expands in the new mental health system, the organization is experiencing a period of significant growth and change, which may differ in important ways from typical organizational expansion. The challenges of scaling up and shifting operations will be informed by the results of this review.

## Aim and objectives of the scoping review

The objectives of this scoping review are to identify key organizational features that facilitate safe inclusion of lived experience in decision-making and practice. The focus is on mental health services that i) are lived experience advocacy or peer support organizations in the mental health sector/s or ii) where lived experience membership (paid or voluntary) is central to advocacy and peer support operations. This scoping review aims to identify:

1. The critical organizational elements of practice, governance, and culture that characterize mental health lived experience organizations, and thus identify potential mechanisms of efficacy.

2. The evidence that exists for the impact of elements of lived experience organizations on members of their workforce.

3. Synthesize forms of evidence, limitations & knowledge gaps, and recommendations in this content area.

## Methods

The review will be guided by the Joanna Briggs Institute (JBI) methodology framework and is being conducted by a multidisciplinary team including lived experience research fellows. This protocol was also developed in line with the and Preferred Reporting Items for Systematic Review and Meta-Analysis Protocols (PRISMA-P) [11]. See S1 File for a complete PRISMA-P checklist for this protocol [12]. The final review will be conducted in line with the scoping review methodology published by Peters [13]. The Preferred Reporting Items for Systematic Reviews and Meta-Analyses Extension for Scoping Reviews (PRISMA-ScR) (www.prisma-statement.org/Extensions/ScopingReviews) will be used to structure the relevant items for reporting the full review, ensuring transparency and reproducibility. This scoping review protocol was registered in Open Science Framework database (registered: July 26, 2022; registration DOI: 10.17605/OSF.IO/NB3S5). The intended commencement and completion dates for this review are July 1, 2022 and April 1, 2023.

### Inclusion criteria

The Population, Concept, and Context framework [14] will be used to determine studies that will be eligible for inclusion.

**Participants.** For the purpose of this review, participants include organizations for lived experience advocacy or peer support in the mental health sector/s. We will include organizations where lived experience membership (paid or voluntary) is central to advocacy and peer support operations. Lived/living experience workforces will refer to peers/consumers or carers who apply their own experience of mental health issues or experience with caring for those with mental health issues to support others in a work context (paid or voluntary).

**Concept.** The concept will comprise the characteristics of the organizations (i.e., elements of organizations that are set up around lived experience and mental illness and successfully support lived experience consumers (e.g., mental health and trauma)). These elements will include organizational practice, governance, and culture that support lived experience workforces. Examples of characteristics will include types of services provided, training and conduct of advocacy work, as well as elements of member support (e.g., recruitment, engagement, training, mentoring/supervision, career progression, trauma-informed interactions, and cultural safety with diverse populations [e.g., First Nations people]).

**Context.** This scoping review will consider any mental health setting, for children, youth, and adults. No geographical restrictions will be applied.

**Types of sources.** This scoping review will include published quantitative, qualitative, and mixed methods data of any study design including primary studies and reviews. Organizational reports, evaluations, and descriptive data on organizational and/or mental health outcomes (e.g., organizational elements and member support) will also be considered. Only references published in English will be considered for inclusion due to time and resource constraints.

### Search strategy

A three-step search strategy will be utilized in this review. An initial limited search of PsycINFO will be undertaken, followed by analysis of the text words contained in the title and

abstract, and of the index terms used to describe the article. A second search using all identified keywords and index terms will be across all included databases. The following databases will be searched: PsycINFO, MEDLINE, CINAHL, ProQuest, and EMBASE. Following recommendations by Aromataris and Riitano [15], unpublished studies and grey literature will also be examined. The unpublished literature search will include ProQuest Dissertations and Theses, Google, Google Scholar, and will also include searching pertinent government and lived experience organization website data to identify white papers (i.e., government reports) and research reports. We will check the reference lists of all included studies and relevant systematic reviews to identify additional studies missed from the original search (for example, unpublished or in-press citations). Studies published in English will be included. Studies published since 2000 will be included as the past two decades has seen considerable growth and advancements in the peer support workforce [16]. See S2 File for the complete MEDLINE search strategy and with explanatory contextual narrative. See S3 File for the complete grey literature search strategy.

Following the search, all identified citations will be uploaded into EndNote V.X9 bibliographic software management program. All duplicates will be removed in Endnote before exporting to Covidence systematic review software [17] for screening. Data will be extracted by two independent reviewers (AV and AM) using data extraction instruments that were predetermined by the research team (see S4 and S5 Files). Depending on the volume of papers that are retrieved, other team members may also be involved in the extraction process. To ensure consistency is met, two reviewers (AV and AM) will pilot test the data extraction instruments by independently charting the data from a select sample (i.e., ≥5 articles). Any disagreements will be resolved through discussion with a third reviewer (JO). Once the pilot-tested data extraction instruments are approved for consistency, data from each included full-text article will be charted by one member and checked by a second member to ensure all relevant information is charted. The draft data extraction tool will be refined as necessary during the process of extracting data from each included paper. If modifications occur, they will be detailed in the review. Where necessary, authors of studies will be contacted to obtain missing information.

Data to be extracted will be separated into two data extraction instruments based on the following themes:

i. Characteristics of included studies (see S4 File)

ii. Effectiveness of CRO elements (see S5 File)

## Critical appraisal of individual sources of evidence

While not necessary for inclusion in a scoping review, a critical appraisal of the references included will be performed using relevant tools from the JBI [18]. One reviewer will rate study risk of bias, with full verification of all judgments (and support statements) by a second reviewer. In addition, the AACODS Checklist will be used to appraise grey literature based on the following criteria: authority, accuracy, coverage, objectivity, date, and significance [19].

## Data management and mapping the results

As recommended by the JBI [18], we will narratively analyze and synthesize the evidence, drawing on lived experience organizational elements to thematically group study findings in the results and discussion sections of the manuscript. The data extracted will be presented in a tabular format using an inductive approach as per Pollock [20] and Pollock [21]. We will analyse the data by quantifying text and completing frequency counts of data extraction items

[14, 18]. The items extracted will address the aims of the scoping review. This analysis will emphasise established and emergent recurring characteristics of organizational practice, governance, and culture to present an overview of prominent features of lived experience organizations according to the literature (see S4 and S5 Files). We are particularly interested in how the data defines organizational structure and values–how the organization is governed and operated and according to which ideological principles. S4 File details further investigation of the specific supports for the organization's membership and workforce, with the expectation that these aspects will be the most unique to lived experience organizations.

We will also be collecting and reporting on the outcomes of the organizations represented in the review to map both how positive outcomes are defined for lived experience organizations and what characteristics are involved in these outcomes. These outcomes will be, where possible, summarised in tables according to S5 File. It is anticipated that outcomes will be significantly varied across organizations and studies and will not be pre-categorised in the manner of the organizational elements and features. Trends and recurring themes observed through mapping organization and study-specific outcomes will be logically summarised and discussed to draw together a clear picture of what the evidence is saying about the impact of these lived experience organizations on their workforce and membership.

To highlight the most and least frequently observed CRO elements, data will be charted in both tabular and graphical form. See S6 File for a tabular template. All information will be contextualised according to S4 File, which defines the study from which the organizational data has been extracted as well as key demographic information about the organizations in question. These reporting methodologies and the tables represented in S4–S6 Files may be further refined once we have the results of the searches, with any modifications being reported in the review.

## Stakeholder engagement

Stakeholders and end users of the scoping review will be involved in the co-creation of this scoping review from review commencement to completion [22]. Stakeholder participation will be to provide essential advice and guidance. Stakeholders will include VMIAC and The Australian Centre for Social Innovation (TACSI). VMIAC and TACSI have regularly consulted with researchers (JM, JO, AV, AM) conducting the review to co-develop all review elements. The development of the review's published and unpublished grey literature search strategies has been conducted by consulting a senior health-science librarian (NP).

## Ethics and dissemination

Ethical approval is not required for this scoping review given it is comprised of publicly available secondary data. Review findings will be disseminated in a peer reviewed journal and shared with lived experience organization stakeholders (i.e., VMIAC and TACSI) through meetings and conferencing.

## Discussion

It is anticipated that this scoping review will map out the current evidence base underpinning best organizational practices in which lived experience workers are involved, particularly in the mental health system. The included evidence will likely highlight tensions between traditional organizational structures and the lived experience workforce and indicate alternate organizing principles and characteristics. Mapping the features of existing organizations will provide an overview of what a lived experience organization looks like according to current evidence, and how this image might develop in a rapidly changing landscape. This broad

interrogation of what has been done to date will be valuable in identifying gaps in current service systems, and opportunities informing further work needs to be done to address them in research, policy, and practice.

This scoping review will map both the organizational features and outcomes reported in the literature, and therefore provide a means of evaluating the best practice approaches amongst the included studies. Where outcomes have been particularly successful and relevant, the characteristics of the organization will be especially useful for researchers and decision-makers moving forward. Significant organizational challenges will also be examined to identify necessary mental health lived experience organization growth areas. The review will act as a critical overview of what policymakers, researchers, and sector-leaders can and should investigate further in the ongoing development of lived experience inclusion and the lived experience workforce. This scoping review will be immediately utilised by associated stakeholders to inform their organizational development and planning work, and it is anticipated that the review will also be of value to the mental health sector at large in informing ongoing work in this emerging and critical aspect of the field.

## Supporting information

**S1 File. PRISMA-P (Preferred Reporting Items for Systematic review and Meta-Analysis Protocols) 2015 checklist.**
(DOCX)

**S2 File. Published MEDLINE database search strategy (searched July 26, 2022).**
(DOCX)

**S3 File. Unpublished and grey literature search strategy.**
(DOCX)

**S4 File. Data extraction instrument 1—Characteristics of included studies separated by study design.**
(DOCX)

**S5 File. Data extraction instrument 2 –Effectiveness of CRO elements.**
(DOCX)

**S6 File. Frequent CRO elements identified.**
(DOCX)

**S7 File. Data extraction instrument 4 –Impact of lived experience organizations on outcomes for members of their workforce.**
(DOCX)

## Acknowledgments

We wish to thank the wider project group and stakeholders at VMIAC and TACSI for their conceptual and contextual contributions.

## Author Contributions

**Conceptualization:** Jessica E. Opie, Jennifer E. McIntosh.

**Methodology:** Jessica E. Opie, Hanan Khalil, Natalie Pearce.

**Project administration:** Jessica E. Opie.

**Supervision:** Hanan Khalil, Jennifer E. McIntosh.

**Writing – original draft:** Jessica E. Opie, An Vuong, Alexandra Macafee.

**Writing – review & editing:** Jessica E. Opie, An Vuong, Alexandra Macafee, Hanan Khalil, Erandathie Jayakody, Christopher Maylea, Jennifer E. McIntosh.

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
