## [Decision Letter · Decision Letter 0]

20 Jan 2023

PONE-D-22-21212Characteristics of Mental Health Lived Experience Workforce Organizations: A Scoping Review ProtocolPLOS ONE

Dear Dr. Opie,

Thank you for submitting your manuscript to PLOS ONE. After careful consideration, we feel that it has merit but does not fully meet PLOS ONE’s publication criteria as it currently stands. Therefore, we invite you to submit a revised version of the manuscript that addresses the points raised during the review process.

We look forward to receiving your revised manuscript.

Kind regards,

Nyanyiwe Masingi Mbeye, Ph.D

Academic Editor

PLOS ONE

Journal Requirements:

2. Please amend your authorship list in your manuscript file to include all authors.

Reviewers' comments:

Reviewer's Responses to Questions

**Comments to the Author**

1. Does the manuscript provide a valid rationale for the proposed study, with clearly identified and justified research questions?

Reviewer #1: Yes

Reviewer #2: Yes

2. Is the protocol technically sound and planned in a manner that will lead to a meaningful outcome and allow testing the stated hypotheses?

Reviewer #1: Partly

Reviewer #2: Yes

3. Is the methodology feasible and described in sufficient detail to allow the work to be replicable?

Reviewer #1: Yes

Reviewer #2: Yes

4. Have the authors described where all data underlying the findings will be made available when the study is complete?

Reviewer #1: No

Reviewer #2: Yes

5. Is the manuscript presented in an intelligible fashion and written in standard English?

Reviewer #1: Yes

Reviewer #2: Yes

6. Review Comments to the Author

You may also provide optional suggestions and comments to authors that they might find helpful in planning their study.

Reviewer #1: The title contains four determinants (mental health, lived experience, workforce, organizations) which make it semantically confusiing. I may suggest "The lived experience workforce: A scoping review from mental health organizations". The literature review should highlight the top features that characterize organizations where lived experience workforce has a true voice rather than capitalizing on how essential it is to have this category of workforce. Moreover, researchers can consider one of the most significant challenges such as power imbalance as the center of the argument; this will help them too in having a well defined search for publications/documents. Furthermore, more details need to be given on the inclusion/exclusion criteria and keep them unchanged all through (line 22- "No restriction on date of publication", page 9, line 7 "Studies published since 2000 will be included". Finally, I suggest the inclusion of the planned statistical tests.

Reviewer #2: 1. This is an important scoping study. The study has already been registered at OSF registries (DOI:10.17605/OSF.IO/NB3S5) where the details of the methods have been described. Data collection began on 1st July and ended 1st December 2022. Rather than publishing the protocol again here I would rather you analyse the data and instead publish the results.

2. Somewhere in the protocol it is stated that "Studies published in English will be included with no restriction on date of publication." However, on page 9 you include the following statement, "Studies published in English will be included. Studies published since 2000 will be included as the past two decades has seen considerable growth .........." Please clarify which period this scoping review will cover.

3. The protocol does not include the key words that will be used to identify the eligible published articles or unpublished reports or policy statements.

4. This scoping review targets to achieve three aims. The 3rd aim "Identify forms of evidence to date and knowledge gaps" is not clear to me. Please clarify this aim further by stating exactly what you intend to do

7. PLOS authors have the option to publish the peer review history of their article (what does this mean?). If published, this will include your full peer review and any attached files.

Reviewer #1: **Yes: **Nariman Ghader

Reviewer #2: **Yes: **Francis Kiweewa

---

## [Author Response · Author response to Decision Letter 0]

24 Jan 2023

To the editorial office: Thank you for arranging this thorough and thoughtful review of our manuscript. We have responded to each comment below, point-by-point.

Reviewer 1, comment 1: The title contains four determinants (mental health, lived experience, workforce, organizations) which make it semantically confusing. I may suggest "The lived experience workforce: A scoping review from mental health organizations". 

Authors’ Reply: We appreciate this feedback and have changed the title of the scoping review to "Characteristics of mental health consumer-run organizations: A scoping review protocol".

Reviewer 1, comment 2: The literature review should highlight the top features that characterize organizations where lived experience workforce has a true voice rather than capitalizing on how essential it is to have this category of workforce. 

Authors’ Reply: We thank the reviewer for this helpful feedback. In response, the study now aims to identify

1. The critical organizational elements of practice, governance, and culture that characterize mental health CROs, and thus identify potential mechanisms of CRO efficacy. 

2. The evidence that exists for the impact of CRO elements on members of their workforce.

3. Synthesize forms of evidence, limitations & knowledge gaps, and recommendations in this content area. 

Reviewer 1, comment 3: Moreover, researchers can consider one of the most significant challenges such as power imbalance as the center of the argument; this will help them too in having a well-defined search for publications/documents. 

Authors’ Reply: We thank the reviewer for this helpful feedback. In response, we will highlight significant challenges as a point of emphasis in the Scoping Review Results and Discussion sections. We have also revised the following text from: 

“This scoping review will map both the organizational features and outcomes reported in the literature, and therefore provide a means of evaluating the best practice approaches amongst the included studies. Where outcomes have been particularly successful (or unsuccessful) and relevant, the characteristics of the organization will be especially useful for researchers and decision-makers moving forward.”

To: “This scoping review will map both the organizational features and outcomes reported in the literature, and therefore provide a means of evaluating the best practice approaches amongst the included studies. Where outcomes have been particularly successful and relevant, the characteristics of the organization will be especially useful for researchers and decision-makers moving forward. Significant organizational challenges will also be examined to identify necessary CRO growth areas.”

Reviewer 1, comment 4: Furthermore, more details need to be given on the inclusion/exclusion criteria and keep them unchanged all through (line 22- "No restriction on date of publication", page 9, line 7 "Studies published since 2000 will be included". 

Authors’ Reply: Thank you for this comment, a publication limitation has been added to the abstract. This now reads: “Included studies will be identified through comprehensive searches of five databases: PsycINFO (Ovid), CINAHL (EBSCO), EMBASE (Ovid), MEDLINE (Ovid), and ProQuest Central. Studies published in English from 2000 onwards will be included”

Reviewer, comment 5: Finally, I suggest the inclusion of the planned statistical tests.

Authors’ Reply: As this is a scoping review, there will be no statistical tests completed. We will narratively synthesize the result, as reported on page 10 and 11 of the manuscript.

Reviewer 2, comment 1: This is an important scoping study. The study has already been registered at OSF registries (DOI:10.17605/OSF.IO/NB3S5) where the details of the methods have been described. Data collection began on 1st July and ended 1st December 2022. Rather than publishing the protocol again here I would rather you analyse the data and instead publish the results.

Authors’ Reply: Thank you for seeing the value in this body of work. We will complete the review proper in the coming months and will then submit that for publication also. 

Reviewer 2, comment 2: Somewhere in the protocol it is stated that "Studies published in English will be included with no restriction on date of publication." However, on page 9 you include the following statement, "Studies published in English will be included. Studies published since 2000 will be included as the past two decades has seen considerable growth .........." Please clarify which period this scoping review will cover.

Authors’ Reply: Thank you for this comment, a publication limitation has been added to the abstract. This now reads: “Included studies will be identified through comprehensive searches of five databases: PsycINFO (Ovid), CINAHL (EBSCO), EMBASE (Ovid), MEDLINE (Ovid), and ProQuest Central. Studies published in English from 2000 onwards will be included”

Reviewer 2, comment 3: The protocol does not include the key words that will be used to identify the eligible published articles or unpublished reports or policy statements.

Authors’ Reply: Thank you for this feedback, in response the key words have been to: “Consumer run organization, lived experience organization, lived experience workforce, mental health, scoping review, protocol”

Reviewer 2, comment 4: This scoping review targets to achieve three aims. The 3rd aim "Identify forms of evidence to date and knowledge gaps" is not clear to me. Please clarify this aim further by stating exactly what you intend to do

Authors’ Reply: Thank you for this comment. The study aims have now been changed to:

1. The critical organizational elements of practice, governance, and culture that characterize mental health CROs, and thus identify potential mechanisms of CRO efficacy. 

2. The evidence that exists for the impact of CRO elements on members of their workforce.

3. Synthesize forms of evidence, limitations & knowledge gaps, and recommendations in this content area. 

Editorial office, comment 1: Please ensure that your manuscript meets PLOS ONE's style requirements, including those for file naming. The PLOS ONE style templates can be found at

Authors’ Reply: The PLOS ONE style templates have been reviewed and the current manuscript adheres to all stylistic requirements. 

Editorial office, comment 2: Please amend your authorship list in your manuscript file to include all authors.

Authors’ Reply: This has now been amended – see cover page document.

Editorial office, Comment 3: Please amend your list of authors on the manuscript to ensure that each author is linked to an affiliation. Authors’ affiliations should reflect the institution where the work was done (if authors moved subsequently, you can also list the new affiliation stating “current affiliation:….” as necessary).

Authors’ Reply: All current author affiliations were the institution were the author worked while completing the review for the review protocol. Author affiliations have now been revised to: 

Jessica E. Opie Ph.D. ab*, An Vuong BPsych ab, Alexandra Macafee BA c , Natalie Pearce BPsySc d, Erandathie Jayakody BCom LLB b, Christropher Maylea DSW b , Hanan Khalil Ph.D. b, Jennifer E. McIntosh Ph.D. ab

a Bouverie Centre La Trobe University, Melbourne, Australia, 3056.

b La Trobe University, Melbourne, Australia, 3000.

c The Victorian Mental Illness Awareness Council (VMIAC), Melbourne, Australia, 3057.

d Library, Latrobe University, Bendigo, Australia, 3551.

Editorial office, comment 4: Please include captions for your Supporting Information files at the end of your manuscript, and update any in-text citations to match accordingly. Please see our Supporting Information guidelines for more information: http://journals.plos.org/plosone/s/supporting-information.

Authors’ Reply: We thank the reviewer for noting this. The manuscript already includes titles for all Supporting information. If a caption is always required this can be identical to the title. 

Editorial office, comment 5: Please review your reference list to ensure that it is complete and correct. If you have cited papers that have been retracted, please include the rationale for doing so in the manuscript text, or remove these references and replace them with relevant current references. Any changes to the reference list should be mentioned in the rebuttal letter that accompanies your revised manuscript. If you need to cite a retracted article, indicate the article’s retracted status in the References list and also include a citation and full reference for the retraction notice.

Authors’ Reply: All manuscript references have been reviewed, and are complete and correct. No retracted paper has been cited in the present manuscript.

---

## [Decision Letter · Decision Letter 1]

20 Feb 2023

PONE-D-22-21212R1Characteristics of mental health consumer-run organizations: A scoping review protocolPLOS ONE

Dear Dr. Opie,

Thank you for submitting your manuscript to PLOS ONE. After careful consideration, we feel that it has merit but does not fully meet PLOS ONE’s publication criteria as it currently stands. Therefore, we invite you to submit a revised version of the manuscript that addresses the points raised during the review process.

We look forward to receiving your revised manuscript.

Kind regards,

Jonas Preposi Cruz

Academic Editor

PLOS ONE

Journal Requirements:

Reviewers' comments:

Reviewer's Responses to Questions

**Comments to the Author**

1. Does the manuscript provide a valid rationale for the proposed study, with clearly identified and justified research questions?

Reviewer #1: Yes

Reviewer #2: Yes

2. Is the protocol technically sound and planned in a manner that will lead to a meaningful outcome and allow testing the stated hypotheses?

Reviewer #1: Yes

Reviewer #2: Yes

3. Is the methodology feasible and described in sufficient detail to allow the work to be replicable?

Reviewer #1: Yes

Reviewer #2: Yes

4. Have the authors described where all data underlying the findings will be made available when the study is complete?

Reviewer #1: No

Reviewer #2: No

5. Is the manuscript presented in an intelligible fashion and written in standard English?

Reviewer #1: Yes

Reviewer #2: Yes

6. Review Comments to the Author

You may also provide optional suggestions and comments to authors that they might find helpful in planning their study.

Reviewer #1: The author has introduced appropriate changes. I still don't see any adjustment to the study title, though. As far as it is for the protocol itself, planned data management is still not fully explained. In response to the author's reply to Reviewer 1- comment 5 and Reviewer 2-comment 1, it is well understood that no statitistical tests will be done in the protocol; however, the manuscript shall fully disclose the researchers' statistical analysis plan that will be implemented at a later stage.

Reviewer #2: The authors have done a great job in addressing the previous review comments and the necessary revisions have made to provide more clarity.

I am satisfied with the responses to the issues I had previously raised.

I have no additional comments.

7. PLOS authors have the option to publish the peer review history of their article (what does this mean?). If published, this will include your full peer review and any attached files.

Reviewer #1: **Yes: **Nariman Ghader

Reviewer #2: **Yes: **Francis Kiweewa

---

## [Author Response · Author response to Decision Letter 1]

21 Feb 2023

To the editorial office: Thank you for arranging this thorough and thoughtful review of our manuscript. We have responded to each comment below, point-by-point.

Comment 1: Does the manuscript provide a valid rationale for the proposed study, with clearly identified and justified research questions?

Authors’ Reply: We are glad the manuscript has met this criterion.

Comment 2: Is the protocol technically sound and planned in a manner that will lead to a meaningful outcome and allow testing the stated hypotheses?

Authors’ Reply: We are glad the manuscript has met this criterion.

Comment 3: Is the methodology feasible and described in sufficient detail to allow the work to be replicable?

Authors’ Reply: We are glad the manuscript has met this criterion.

Comment 4: Have the authors described where all data underlying the findings will be made available when the study is complete?

Authors’ Reply: In response, the following text has been included on page 14 of the manuscript:

“Data availability statement: All relevant data from this study will be made available upon study completion.”

Comment 5: Is the manuscript presented in an intelligible fashion and written in standard English?

Authors’ Reply: We are glad the manuscript has met this criterion.

Comment 6: 

Reviewer #1: The author has introduced appropriate changes. I still don't see any adjustment to the study title, though. As far as it is for the protocol itself, planned data management is still not fully explained. In response to the author's reply to Reviewer 1- comment 5 and Reviewer 2-comment 1, it is well understood that no statistical tests will be done in the protocol; however, the manuscript shall fully disclose the researchers' statistical analysis plan that will be implemented at a later stage.

Authors’ Reply: Thank you for this feedback. We have changed the title from “Characteristics of mental health consumer-run organizations: A scoping review protocol” to “Practice, governance, and culture characteristics of lived experience organisations, and evidence of efficacy: A Scoping Review Protocol” 

Regarding the data management plan, scoping reviews are a type of evidence synthesis that aim at identifying and mapping the literature on a particular topic within a specific context. Unlike, systematic reviews, they do not have any statistical analysis due to the heterogeneity of the types of studies included. Data synthesis includes a narrative synthesis of the results to address the research question as per Pollock et al. (2022a) and Pollock et al. (2022b).

We have included the following paragraph on page 11:

“The data extracted will be presented in a tabular format using an inductive approach as per Pollock et al. (2022a) and Pollock et al. (2022b). We will analyse the data by quantifying text and completing frequency counts of data extraction items (Aromataris & Munn, 2020; Peters et al., 2022). The items extracted will address the aims of the scoping review. This analysis will emphasise established and emergent recurring characteristics of organizational practice, governance, and culture to present an overview of prominent features of lived experience organizations according to the literature (see Supporting Information 4).”

References:

Aromataris E, Munn Z, editors. JBI manual for evidence synthesis. Adelaide: The Joanna Briggs Institute; 2020. http://dx.doi.org/10.46658/ JBIMES-20-01 

Peters, M. D., Godfrey, C., McInerney, P., Khalil, H., Larsen, P., Marnie, C., ..., & Munn, Z. (2022). Best practice guidance and reporting items for the development of scoping review protocols. JBI Evid Synth, 20(4), 953-968. 

Pollock D, Tricco AC, Peters MD, Mclnerney PA, Khalil H, Godfrey CM, Alexander LA, Munn Z. Methodological quality, guidance, and tools in scoping reviews: a scoping review protocol. JBI Evidence Synthesis. 2022a Apr 1;20(4):1098-105 

Pollock D, Peters MD, Khalil H, McInerney P, Alexander L, Tricco AC, Evans C, de Moraes ÉB, Godfrey CM, Pieper D, Saran A. Recommendations for the extraction, analysis, and presentation of results in scoping reviews. JBI evidence synthesis. 2022b Sep 8:10-1124.

Reviewer #2: The authors have done a great job in addressing the previous review comments and the necessary revisions have made to provide more clarity. I am satisfied with the responses to the issues I had previously raised. I have no additional comments.

Authors’ Reply: We thank you kindly for this feedback and for taking the time to review this paper.

---

## [Editor Report · Decision Letter 2]

6 Mar 2023

Practice, governance, and culture characteristics of lived experience organizations, and evidence of efficacy: A scoping review protocol

PONE-D-22-21212R2

Dear Dr. Opie,

We’re pleased to inform you that your manuscript has been judged scientifically suitable for publication and will be formally accepted for publication once it meets all outstanding technical requirements.

Kind regards,

Jonas Preposi Cruz

Academic Editor

PLOS ONE